# Photovoltaic modules evaluation and dry-season energy yield prediction model for NEM in Malaysia

Syed Zahurul Islam[1]☯*, Mohammad Lutfi Othman[2‡]*, Muhammad Saufi[1‡], Rosli Omar[1‡], Arash Toudeshki[3☯], Syed Zahidul Islam[4‡]

1 Faculty of Electrical & Electronic Engineering, Universiti Tun Hussein Onn Malaysia(UTHM), Parit Raja, Johor, Malaysia, 2 Department of Electrical and Electronics Engineering, Advanced Lightning, Power and Energy Research (ALPER), Faculty of Engineering, Universiti Putra Malaysia, Serdang, Selangor, Malaysia, 3 School of Engineering, University of California, Merced, CA, United States of America, 4 Radiation Solutions Inc, Mississauga, ON, Canada

☯ These authors contributed equally to this work.
‡ These authors also contributed equally to this work.
* zahurul@uthm.edu.my (SZI); lutfi@upm.edu.my (MLO)

**Data Availability Statement:** All relevant data are within the manuscript and its Supporting Information files.

## Abstract

This study analyzes the performance of two PV modules, amorphous silicon (a-Si) and crystalline silicon (c-Si) and predicts energy yield, which can be seen as facilitation to achieve the target of 35% reduction of greenhouse gases emission by 2030. Malaysia Energy Commission recommends crystalline PV modules for net energy metering (NEM), but the climate regime is a concern for output power and efficiency. Based on rainfall and irradiance data, this study aims to categorize the climate of peninsular Malaysia into rainy and dry seasons; and then the performance of the two modules are evaluated under the dry season. A new mathematical model is developed to predict energy yield and the results are validated through experimental and systematic error analysis. The parameters are collected using a self-developed ZigBeePRO-based wireless system with the rate of 3 samples/min over a period of five days. The results unveil that efficiency is inversely proportional to the irradiance due to negative temperature coefficient for crystalline modules. For this phenomenon, efficiency of c-Si (9.8%) is found always higher than a-Si (3.5%). However, a-Si shows better shadow tolerance compared to c-Si, observed from a lesser decrease rate in efficiency of the former with the increase in irradiance. Due to better spectrum response and temperature coefficient, a-Si shows greater performance on output power efficiency (OPE), performance ratio (PR), and yield factor. From the regression analysis, it is found that the coefficient of determination ($R^2$) is between 0.7179 and 0.9611. The energy from the proposed model indicates that a-Si yields 15.07% higher kWh than c-Si when luminance for recorded days is 70% medium and 30% high. This study is important to determine the highest percentage of energy yield and to get faster NEM payback period, where as of now, there is no such model to indicate seasonal energy yield in Malaysia.

**Funding:** Syed Zahurul Islam, Research Management Center, Research Fund E15501, Universiti Tun Hussein Onn Malaysia (UTHM), www.uthm.edu.my. Mohammad Lutfi, 9671700, Geran Putra Berimpak, University Putra Malaysia (UPM), www.upm.edu.my. The funders had no role in study design, data collection and analysis, decision to publish, or preparation of the manuscript. Syed Zahidul Islam is an employee of the commercial Canadian company, Radiation Solutions Inc. The funders had no role in study design, data collection and analysis, decision to publish, or preparation of the manuscript. The specific roles of these authors are articulated in the 'author contributions' section.

**Competing interests:** Syed Zahidul Islam is an employee of the commercial Canadian company, Radiation Solutions Inc. There are no patents, products in development or marketed products to declare. This does not alter our adherence to PLOS ONE policies on sharing data and materials.

**Abbreviations: Nomenclature:** *A*, Area ($m^2$); a–Si, Amorphous Silicon; *b*, Number of collected data from 08:30 to 17:30; c–Si, Crystalline Silicon; *D*, Total number of days in the dry season; *E*, Energy yield (kWh); FF, Fill factor; *H*, Humidity (%); *I*, Current (A); MAPE, Mean absolute percentage error; MBE, Mean bias error; *N*, Number of packets; NEM, Net energy metering; OPE, Output power efficiency (%); *P*, Power (W); PR, Performance ratio; PV, Photovoltaic; R$^2$, Coefficient of determination (R–squared); RMSE, Root mean square error; *r*, Coefficient of correlation; *S*, Size of SD card (bit); *Sh*, Sun-hour (h); SMAPE, Symmetric mean absolute percentage error; STC, Standard test condition; *T*, Temperature (˚C); $T_A$, Ambient temperature (˚C); $t_c$, Critical t-statistic; $t_s$, t-statistic; *V*, Voltage (V); *Z*, Solar irradiance ($\frac{W}{m^2}$); *α*, Probability of *medium luminance day*; *β*, Probability of *high luminance day*; *ε*, Temperature coefficient (˚C$^{-1}$); *η*, Module efficiency (%); *φ*, Collected each environmental or electrical data; **Subscripts:** *a*, measured/actual; *avg*, average; *e*, estimated; *M*, module; *max*, maximum; *oc*, open circuit; *STC*, at STC condition; *sc*, short circuit.

# Introduction

To lessen the effect on human life and emancipate environment from crippling by exhaling carbon and other greenhouse gases, solar energy is one of the major alternative energy harvesting systems for generating electricity [1, 2]. Malaysia is one of the tropical countries comprising of two regions; Peninsular West and East Malaysia with tremendous solar potential (22–24 and 14–24 $MJ/m^2/day$ respectively for generating electricity [3]. This could meet its projected electricity peak-demand of 23.099 GW in 2019 which is reflecting 39.47% higher than the peak-demand in 2013 [4]. However, there are short and long term climate challenges in Peninsular West Malaysia that pose threat to electricity generation from solar [5]. Short term effects are intermittent cloud and supply disruption where long term effects are high ambient temperature, humidity, and Southeast Asian haze pollution, and extreme rainfall [6, 7].

In the 10$^{th}$ Malaysia plan, crystalline type PV modules were widely used due to their attractive efficiencies and it is promoted intensively in the 11$^{th}$ plan (2016–2020) through NEM implementation [8]. The efficiencies of the PV modules are specified by the manufacturer in standard test condition (STC) defined as 1000 $\frac{W}{m^2}$ incident irradiance, 25˚C module temperature, and 1.5 air mass. However, PV module efficiency in STC is not applicable for Malaysia climate condition since 33˚C ambient temperature can significantly affect the open circuit voltage by −104 $\frac{mV}{˚C}$ of the PV [9, 10]. This can reduce 0.15% of FF and 0.4 ∼ 0.5% of maximum output power, for every 1 ˚C increase in module temperature [11].

The performance of different PV modules varies from STC measurement and it depends on geographical position and climatic condition. Based on Malaysia's real climate variation, there should be an analysis on performance of Malaysian Energy Commission recommended PV modules and its energy yield modeling for the net energy metering (NEM, previously called FiT). Seasonal based performance and energy yield model of the recommended PV modules due to climate regime in Malaysia are still intangible. In this study, we have evaluated the electrical performance of the two PV modules, namely c-Si and a-Si for the case of peninsular Malaysia during the dry season. The performance parameters are module efficiency, output power efficiency (OPE), performance ratio (PR), fill factor (FF), energy yield, and yield factor. From the evaluation, we have developed a model which predicts the dry season's energy yield of the modules. The outcome of the research can be seen as a support of the 11$^{th}$ Malaysia Plan (2016-20) development – accelerate renewable energy capacity in NEM as well as achieve target of 35% reduction of greenhouse gas emissions by 2030. The novelty of this study is that we have conducted regression analysis on a range of environmental and electrical parameters to investigate their degree of relationship under the dry season, while predicting the two modules' energy-yield as a part of payback investment in NEM.

To rev-up 20% green energy by 2025, Malaysia Energy Commission has taken many initiatives and policies through establishing large scale solar generations [12]. Some of them are, namely 197 MW Quantum solar park and 65 MW Jasin solar plant. However, Malaysia is blessed with 62.3% of tropical forests containing rich flora of animal species [13]. The alteration of it by large solar plants would lead to the disturbance of the natural ecosystem. This would alter forests topology, crop yields, water supplies which might eventually lead to famine. Many plants and animal species would be threatened, and some would likely become extinct, for instance, Sumatran rhinoceros is one of the extreme rare species in Malaysia. Therefore, it would be irrational to alter the forest topology by the large solar generations.

For this mutual exclusive challenge in Malaysia perspective, one of the best alternatives could be the roof–top photovoltaic (PV) system that is also supported by the Malaysian government. The NEM has been rolled–out in Malaysia since December 2011 which obliges the distribution licensee in Peninsular West Malaysia, Tenaga Nasional Berhad to purchase from

the approved applicants, the electricity produced from indigenous renewable resources at a fixed price and duration. Due to the encouragement of the Malaysian Government on NEM, recent trend shows increased number of total NEM generation, from 31.6 to 362.2 MW between 2012 and 2017 [2, 12].

Considering the outdoor real weather condition, many researchers have conducted experiment to scrutinize the actual performance of different types of PV modules. The outcomes of the PV modules at different regions including Malaysia have been published in the literature, as shown in the synopsis in Table 1. In view of all that has been mentioned in the peer-reviewed literature, there is no study on seasonal categorization from meteorological data analysis and energy yield model for the NEM payback in Malaysia. Previous studies in Malaysia are limited to performance analysis with seasonal categorization, regression analysis, energy modeling, and validation. Most of the researches conducted in Malaysia consider the climate as 'tropical' without any categorization. The data collected on specific days depict the performance of the PV based on that particular weather. Due to that, two researchers found different result in terms of module efficiency, such as c-Si and poly crystalline are found to be highest module by [15] and [16] respectively. A multiple regression model was predicted for output power by [7], however the key environmental parameter was the dust thickness on the PV surface due to the Southeast Asian haze in 2013. Some researchers have analyzed the performance of the modules for tracking system, finding optimum tilt angle, and cell design under desert climate [26], but these are not directly related to our NEM study. Researchers from other countries, such as Pakistan [19, 21], Colombia [22], Australia [14], Southeast UK [18], Doha [17] etc. also conducted similar analysis of the different modules. In these studies, performance is also measured for their distinct environmental parameters, inter row spacing, and dust on the surface of the module.

As part of both the environmental and electrical data collection, most of the researchers have considered data logging methods, such as environmental sensors or pyranometer integration to computer via wired connection and digital multimeter or solar simulator [15, 16, 18, 19, 27]. However, few effective methods, such as real–time digital simulator-based novel system [28], high–speed four–channel digital oscilloscope [29], Façade technology [24] and automated measurement system [30] were considered by some researchers for performance test and analysing the PV.

In our study, electrical and environmental parameters are recorded using solar analyser and self-developed ZigBeePRO-based smart wireless communication system respectively. Prior to implement our system, a mathematical model is developed to ensure all the environmental data to be accommodated in to a 2GB memory for at least one experimental day. The latest ZigBeePRO with Waspmote microcontroller and smart metering board used in this study is convenient for sensor integration, longer coverage support, low power consumption, large number of child node integration, and better data encryption over Wi-Fi [2]. ZigBeePRO is recommended in this research as a wireless sensor network because it offers additional features over the other wireless transmission protocols as well as ZigBee. Commercially available ZigBeePRO range can go up to 7km, line of sight ([31]), much higher than other wireless transmission protocols, such as WiFi (100m or more), Bluetooth (1-100m), and ZigBee (10-100m). It is also superior to other networks in terms of guaranteed data transmission capability and automatic detection of the addition or absence of nodes, without any manual intervention. In addition, ZigBeePRO protocol supports more than 65000 nodes with extended battery life compared to either WiFi (>1000 nodes) or Bluetooth (7 nodes). An extensive discussion on the most influential feature of ZigBeePRO for distributed solar energy monitoring as applied to the field of smart grid can be found in these authors' works [2, 32, 33].

The contribution of this research can be summarized in three folds:

**Table 1. Summary of major past studies on PV performance evaluation conducted in Malaysia and other countries.**

| Year | Ref. | Location, Climate & Setup | Significant Outcomes | Remarks/Different from this study |
|------|------|---------------------------|----------------------|-----------------------------------|
| 2004 | [14] | Perth, Australia, all, 13-19months, c-Si(75), LGBC c-Si (85), SX-75 p-Si(75), PW750/70 p-Si (70), 3j a-Si (64), and CIS (40) | • a-Si produces 15% (summer) and 8% (winter) more energy compared with c-Si.<br>• CIS module is higher energy producer (between 9-13%) than c-Si due to its higher temperature coefficient. | • Performance analysis of 6 types of modules<br>• Average ambient temperature is 16.5-28°C, much lower than Malaysia<br>• No modeling or regression analysis |
| 2009 | [15] | Bangi, Malaysia, hot-sunny, 3 days (moderate, cloudy, sunny), a-Si (64), c-Si(75), mc-Si(65), CIS(40) | • c-Si and multicrystalline (ms-Si) performance are found to be better than CIS and a-Si<br>• CIS and a-Si relatively show better performance than c-Si and mc-Si when cloudy climate<br>• c-Si is found to be highest efficient module<br>• 3 days' average efficiencies of a-Si, mc-Si, CIS, and c-Si are 2.23, 5.14, 3.99, and 6.87% respectively | • Mainly performance analysis of 4 types of modules<br>• No info on experimental month and sun-hour<br>• No regression analysis or modeling on energy yield |
| 2012 | [16] | Pinang island, Malaysia, dry, 4 days, mono and poly crystalline(NA), a-Si(NA), single axis solar tracker | • Poly crystalline is found to be high efficient module (7.97%)<br>• a-Si attains high output power | • Tracker is not applicable for NEM<br>• Performance analysis is not detail<br>• No module specification<br>• No modeling or significant analysis |
| 2013 | [17] | Doha, Qatar, desert, NA, c-Si(120), a-Si (100) | • a-Si is more sensitive to temperature and humidity but more robust against tiny dust particles than c-Si | • Limited environmental parameters<br>• Performance analysis is based on dust, temperature, and humidity |
| 2014 | [18] | Brighton, Southeast UK, all, 1 year, mono crystalline (10kW roof-top) | • Small fine particles can cause 11% less light transmittance to the fixed flat type module<br>• Transmittance is linear with tilt angle | • Performance analysis is based on dust and tilt angle.<br>• Different climate than Malaysia |
| 2014 | [19] | Taxila, Pakistan, winter, 45 days, c-Si (45), p-Si (40), and a-Si (40) | • c-Si is the highest efficienct module (13.01)% among all<br>• a-Si possesses the highest average PR | • Only performance is evaluated<br>• No regression analysis or modeling |
| 2015 | [7] | Serdang, Malaysia, hazy, 30 days, mono-crystalline (1kW) | • Degradation is about 41.84% in output power and 10% in efficiency during the Southeast Asian haze pollution, 2013 | • Performance is measured based on dust and haze<br>• Regression analysis are for predicting output power only<br>• Models are not validated |
| 2016 | [20] | Pekan, Malaysia, NA, 31 days, multicrystalline (5kW grid connected) | • Propose PV model based on three electrical parameters, namely photo-current, reverse diode saturation current, and ideality factor of diode<br>• Model is validated through experimental data and compared with other studies | • Only one type of PV is considered<br>• No seasonal categorization<br>• Consider 1 month data as a reference for whole year<br>• Only 3 environmental parameters are take into account, such as ambient and module temperature, and solar irradiance<br>• No further analysis on PR, OPE energy yield, and yield factor<br>• No model on output power or energy yield |
| 2019 | [21] | Bahawalpur, Pakistan, desert, 1 year, poly crystalline (two similar 100MW plant adjacent to each other) | • Average annual difference is 4%<br>• Approve and proper design may increase energy of US$ 0.85 million per year | • Concern is to find factors for annual degradation rate<br>• The factors are inter row spacing, tilt angle, negative temperature coefficient of power<br>• Evaluation is based on the factors<br>• No analysis for environmental parameters |
| 2019 | [22] | Medellin, Colombia, ambient temperature 18–42°C and irradiance 0–1200 $\frac{W}{m^2}$ 500 h, Perovskite and silicon module (NA) | • Linear relationship is to be found between power and short circuit current<br>• Open circuit voltage of perovskite is nonlinear and shows better performance with temperature at high irradiance | • Different module type and region<br>• No modeling<br>• PV capacity is not defined |

*(Continued)*

**Table 1.** (Continued)

| Year | Ref. | Location, Climate & Setup | Significant Outcomes | Remarks/Different from this study |
|------|------|---------------------------|----------------------|-----------------------------------|
| 2019 | [23] | Ipoh, Malaysia, NA, dye-sensitised, simulation using 'SimaPro' | • Efficiency and irradiance are inversely proportional<br>• Cumulative energy demand is $18.75 \frac{GJ}{kWh}$<br>• Greenhouse gas emission rate is $70.52 \frac{gCO_{2-eq}}{kWh}$. | • Only 3 environmental indicators are analyzed<br>• There are cumulative energy demand, energy payback time, greenhouse gas emission rate |
| 2019 | [24] | Ulster University, Northern Ireland, 20-100 h, 600-800 $\frac{W}{m^2}$, hybrid PV thermal, indoor simulation | • Overall heat retention efficiency of hybrid PV solar thermal is 65% | • Only indoor experiment<br>• Mainly thermal performance is analysed<br>• Actual environmental parameters are not considered |
| 2019 | [25] | Seoul, South Korea, cold, 730 days, c-Si (260) | • Humidity is found significant in prediction model at low irradiance, low ambient temperature, and high humid | • 6 prediction models on output power<br>• Only root mean square and mean absolute percentage error are calculated<br>• Mainly cold climate, annual average temperature is 10–15°C, different than climate of Malaysia |

Sources are from 2004-2019.

- Analysis of 63 years meteorological rainfall data in peninsular Malaysia where the dry season is chosen for conducting the performance analysis of the PV modules (c–Si and a–Si). The effectiveness of environmental data collection is ensured by a self-developed ZigBeePRO–based smart wireless communication system in an aim of obtaining the data at higher frequency of 3 samples/minute.

- The prediction of the PV modules' performance in terms of energy yield in kWh, that is, the deviation from the STC stated by the modules' manufacturers, is modelled in a manner analogous to the NEM system for the dry climate condition.

- The performance of PV modules is modeled by regression relationship between the environmental and electrical parameters with stochastic analysis. The relationship is evaluated by determining significant statistical indicators, namely, coefficient of correlation ($r$), coefficient of determination ($R^2$), mean bias error (MBE), root mean square error (RMSE), mean absolute percentage error (MAPE), and symmetric mean absolute percentage error (SMAPE).

The organization of this paper is as follows. Section 1 presents the analysis of meteorological data for categorizing peninsular west Malaysia climate. It also highlights vernal and solstice factors for positioning solar module. Then in section 2, hardware setup for electrical parameters using the ZigBeePRO-based smart wireless communication system is explained. Section 3 shows the a–Si and c–Si modules' performance evaluation in three different perspectives. Then regression and statistical analysis on the modules' performance and its validation have been included in this section. This section also explains the estimation model for energy yield in NEM with validation. Finally, section 4 concludes the overall outcome of this research.

## 1 Overview of Peninsular Malaysia climate

In Peninsular Malaysia, the average day–time ambient temperature is 33°C, humidity of 80–90% other than dry season, average cloud-covered factor of 6.5 [1], and average 135.285–366.985 mm rainfall [34]. According to the Malaysian Meteorological department data between 1951 and 2018, three main types of seasonal variation are observed in peninsular west

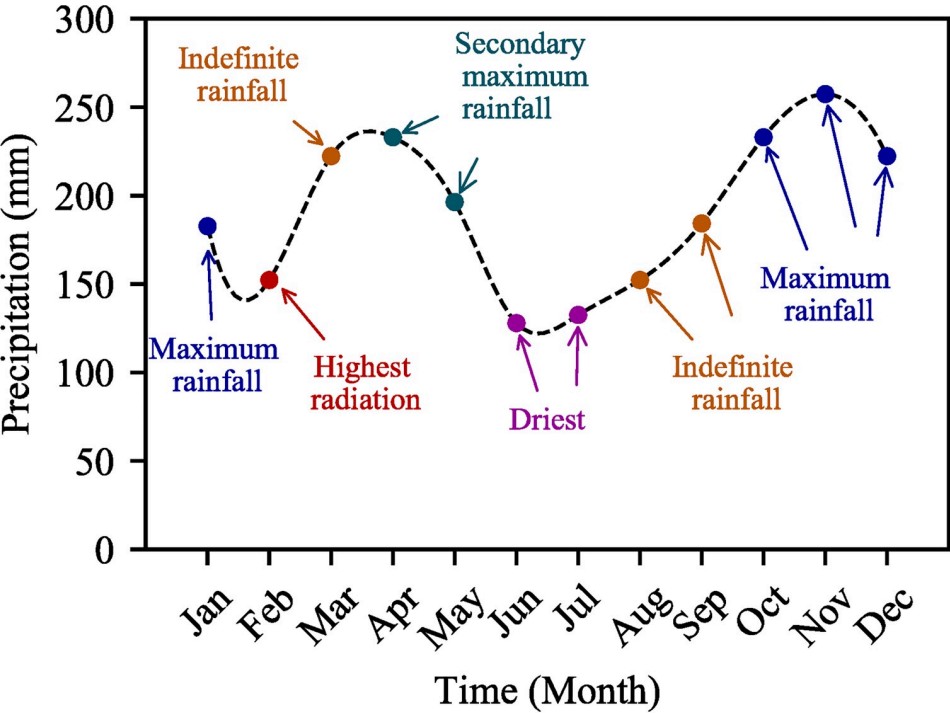

**Fig 1. Seasonal variation of rainfall in Peninsular Malaysia.**

Malaysia: maximum rainfall, secondary maximum rainfall, and the dry season (shown in Fig 1) [35, 36]. In this peninsular, maximum rainfall occurs in the months of October, November, December, and January; however, southwest region of the peninsular has recorded extreme rainfall during October and November (*e.g.*, the heaviest rainfall 9–11 December 2004, 600 $\frac{mm}{day}$ [34, 37, 38]). The secondary maximum rainfall is recorded in April and May. The trend of rainfall since 1951 in the southwest peninsular is linearly increasing by year, rainfall (mm) = 7.0458 × year + 2036.1 [35]. According to the standardized precipitation index or SPI, the prolonged dry months are June and July where the least rainfall is observed, for example, total rainfall received in June 2015 is less than 100 mm. Another category is indefinite rainfall within 200–300 mm in the months of March, August, and September. The highest solar radiation is achieved within the period of February–March [35]. Furthermore, peninsular Malaysia sky is mostly cloudy, 80% of days in a year, thus plummets substantial solar irradiance [1]. However, the sky is generally clearer in the mornings and cloudy in the afternoons. During the rainy seasons, rainfalls are experienced between 14:45 and 18:00, averagely. This means, the harvested solar irradiance during afternoon time should be significantly less than the irradiance during morning time, with the same angular position of the sun. On the other hand, no or fewer rainfall days are generally observed during the dry season.

Based on observations from the meteorological data, each seasonal category is consisting of similar indices, such as solar irradiance, rain/no–rain, cloud factor, and humid level. Our observation is also supported by all the previous researchers where they state the climate of Malaysia as predictable weather, hot and humid all year round, and no large variation in temperature [12, 15]. A research from the analysis of 10 years meteorological data shows that average solar irradiance of June and July is approximately same. It also describes very mere difference in ambient temperature in June (28°C) and July (27.7°C) [39]. Similarly, another

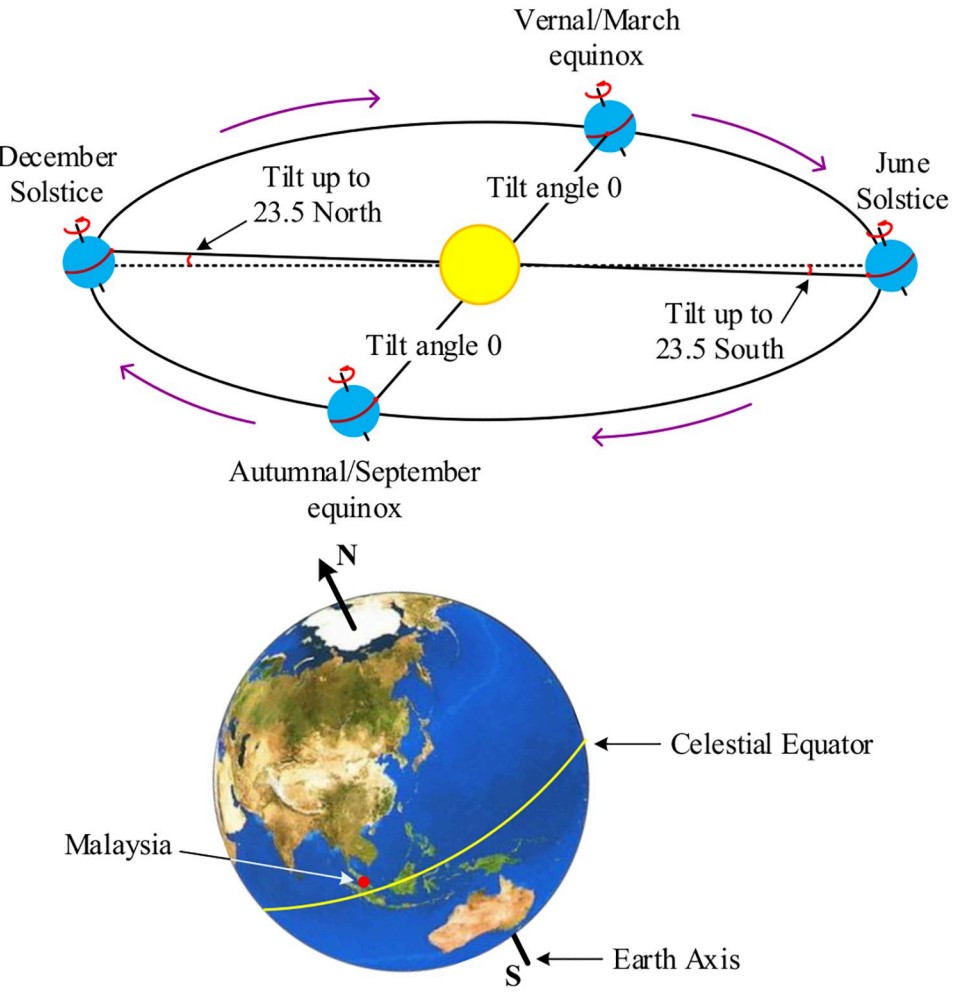

**Fig 2. Vernal/March equinox occurs when the sun directly shines the celestial equator.** This also happens in autumnal/September equinox. On both equinox days, tilt angle is 0˚. Other days of the year, the earth axis is tilted at an angle of approximately 23.5˚ with respect to the eclipse on both solstice days. Reprinted from [43] under a CC BY license, with permission from UPM, original copyright 2016.

research describes Malaysia as 'mere distinctive season country and its climate is hot and humid' [40]. Due to the similar indices, we considered 5 days' of dry climate data to estimate seasonal–based energy yield for peninsular Malaysia.

One of the key points of harvesting maximum solar irradiance is when the panels are perpendicular to the sun rays. For the best performance, solar tracker can enhance the PV efficiency by a factor of 40–48% [41]. However, installation of solar tracker in NEM system is neither cost effective nor feasible for small scale capacity. Another key point is the position of the sun that varies throughout the year. It makes an angle of up to 23.5˚ with respect to the equator towards the north in one half of the year, whereas this angle is tilted towards the south in the other half (described in Fig 2) [42]. Therefore, for one sided PV panel installation on the roof top under NEM system, it is not possible for the panels to achieve the maximum output in a year. To overcome this problem, both northern and southern sided-panel can be installed on the roof top in order to harvest maximum solar irradiance.

## 2 Hardware setup for electrical parameter acquisition

Efficiency, OPE, PR, FF, and yield factor are essential key indices to evaluate the performance of a PV module. These key indices can be obtained from the model equations where the variables are electrical and environmental parameters. The electrical parameters ($V_{max}$, $V_{oc}$, $I_{max}$, $I_{sc}$) are measured using solar analyser (Prova 200) and environmental parameters (solar irradiance, module and ambient temperature, humidity, and wind speed) are acquired by self–developed ZigBeePRO–based smart wireless communication system. The required model equations are explained as follows.

The module efficiency ($\eta$) can be obtained using Eq (1) [15].

$$\eta = \frac{P_a}{Z_a \times A_m} = \frac{I_a \times V_a}{Z_a \times A_m} \times 100 \tag{1}$$

The other parameter indices, such as OPE and PR are obtained by Eqs (2) and (3) respectively.

$$OPE = \frac{P_a}{P_{max,\text{STC}}} \times 100 \tag{2}$$

$$PR = \frac{P_a \times Z_a}{P_{max,\text{STC}} \times Z_{\text{STC}}} \tag{3}$$

FF can be determined by considering maximum power, short circuit current, and open circuit voltage of a PV module, shown in Eq (4).

$$FF = \frac{V_{max} \times I_{max}}{V_{oc} \times I_{sc}} \tag{4}$$

Yield factor can be determined by Eq 5.

$$\text{Yield factor} = \frac{P_a \times h}{P_{max,\text{STC}}} \tag{5}$$

where $P_a$ is the measured actual power (W); $P_{max,\text{STC}}$ is the maximum power in STC (W); $I_a$ is the measured actual current (A); $I_{max}$ is the maximum current (A); $V_a$ is the measured actual voltage (V); $V_{max}$ is the maximum voltage (V); $V_{oc}$ is the open circuit voltage (V); $I_{sc}$ is the short circuit current (A); $Z_{\text{STC}}$ is the solar irradiance in STC ($\frac{\text{W}}{\text{m}^2}$); $Z_a$ is the measured actual solar irradiance ($\frac{\text{W}}{\text{m}^2}$); and $A_M$ is the area of the module ($m^2$).

We have considered two popular commercially available PV modules (c–Si and a–Si) where the specification in STC is given in Table 2. The cost of PV modules is region–specific and varies greatly depending on the market; however, it has been declined gradually in recent years. For the world market, the up–to–date PV module price is US$0.736/Wp [44]. In this study, the cost of the PV modules is US$3.5/W (c–Si) and US$1.75/W (a–Si) according to the supplier price quotation.

**Table 2. Specification of c–Si and a–Si PV modules.**

| Type | Size (mm) | $V_{max}$(V) | $I_{max}$(A) | $V_{oc}$(V) | $I_{sc}$(A) | $P_{max}$(W) | $\eta$(%) | Manufacturer | Cost(US$) |
|---|---|---|---|---|---|---|---|---|---|
| c–Si | 493 × 315 | 17.4 | 1.14 | 21.7 | 1.22 | 20 | 12.9 | Libelium(MSOLAR) | 3.5/W |
| a–Si | 292 × 142 | 17 | 0.10 | 21 | 0.13 | 1.7 | 4.0 | Solar voltaic | 1.75/W |

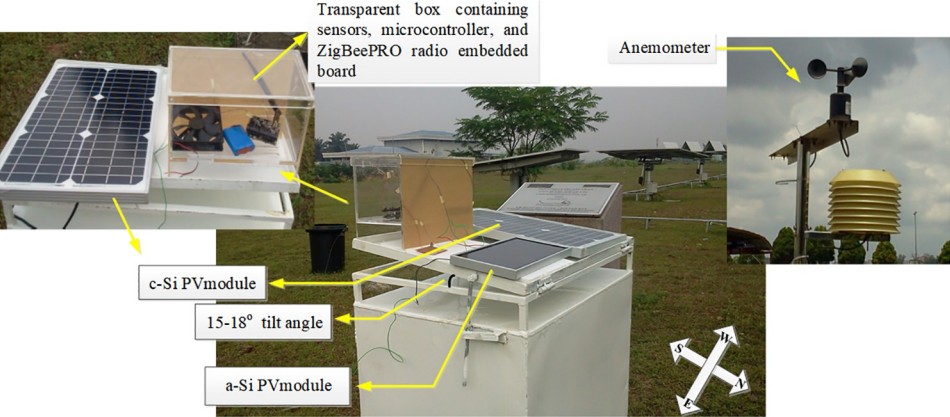

**Fig 3. Outdoor electrical and environmental data collection setup for a–Si and c–Si module.** Location is at UPM solar farm, coordinate 22.945˚ North and 101.75˚ East. 15-18˚ tilt angle is maintained to install the modules on a closed–rack type roof-top facing the north. This direction makes the modules cooler by the blowing wind, from east to west. Transparent box contains ZigBeePRO distribution node consisted of environmental parameter measurement sensors, embedded board, and communication radio. Thermocouples measure the ambient and the modules' temperature. Humidity and luminosity sensors measure the humidity and the solar irradiance respectively. Anemometer is installed separately for measuring wind speed. Reprinted from [43] under a CC BY license, with permission from UPM, original copyright 2016.

During the dry months (Jun–Jul), no rain was observed at Klang valley region, southwest peninsular Malaysia where experimental data was collected. The days considered for the experiment were $12^{th}$, $15^{th}$, $16^{th}$, $19^{th}$, and $20^{th}$ of July corresponding to day1 to day5, respectively. Both modules were installed on fixed roof closed–rack at tilt angle of 15˚ (In Malaysia, 15˚ optimum tilt angle is found by [45]) without considering any sun tracker. Fig 3 shows the outdoor experimental setup located at UPM solar farm, coordinate 2.945˚ North and 101.75˚ East.

The ZigBeePRO-based smart wireless communication system is illustrated in Fig 4. The technique has been adopted from the previous works of these authors [2, 32, 46]. Here in brief, temperature, humidity, and luminosity sensors were interfaced with smart metering and microcontroller board (combining embedded board) with ZigBeePRO communication radio. The temperature sensor MCP9700A is connected to pin6 of the smart metering board for reading analog temperature of the PV module. The other three parameters, such as ambient temperature, humidity, and solar irradiance are measured using identical temperature sensor (MCP9700A), humidity sensor (808H5V5), and luminosity sensor (TSL2561). The sensors specifications are shown in Table 3, Appendix. For simplicity, an approximate conversion of $0.0079 \frac{W}{m^2}$ per Lux is considered. All the sensors are accommodated within the smart metering board which is interfaced with Waspmote microcontroller board. All the sensors are manufacturer-calibrated. Additionally, a 2 GB micro SD card for data recording and a ZigBeePRO radio are interfaced to the embedded board for transferring data to the control centre through the ZigBeePRO gateway.

The data collection was conducted through remote data monitoring system saving environmental parameters to the SD card and simultaneously sending the data to the control centre using ZigBeePRO communication in every 20 second. The 20 seconds interval ensures that all the data is accommodated for at least one experimental day into the 2 GB SD card by the

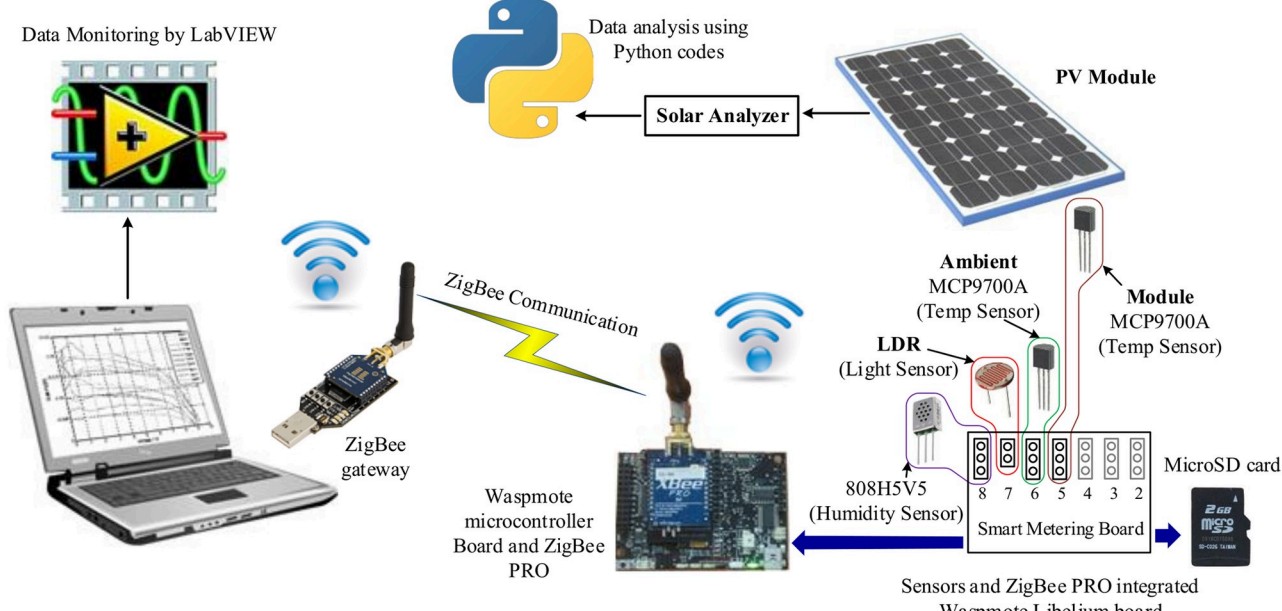

**Fig 4. Integration of sensors, embedded board, and communication module.** Sensors: thermocouples, luminosity or LDR, and humidity. Embedded board: refers to the microcontroller and smart metering board. ZigBeePRO: communication module. Micro SD: attached to embedded board for storing sensors data. Solar Analyzer: retrieved four electrical data, such as open circuit voltage, short circuit current, max voltage and max current of PV module. ZigBeePRO gateway: installed at the control centre for data acquision. LabVIEW program: monitoring SD card data from the control centre. Reprinted from [43] under a CC BY license, with permission from UPM, original copyright 2016.

mathematical relationship in Eq (6).

$$N = \frac{\text{data rate (bit/s)} \times 32400(\text{s})}{\text{packet size (bit)}} \tag{6}$$

Here, *N* is the total number of packet during 9 hours experimental time of a day (without any delay). Based on the ZigBeePRO specification [31], the parameters of Eq (6) can be set: packet size = 1280 bits with header and checksum

**Table 3. Sensors specification.**

| | Humidity Sensor | Temperature Sensor | Luminosity Sensor |
|---|---|---|---|
| Sensor Model | 808H5V5 | MCP9700A | TSL2561 |
| Measuring Range | 0 to 100%RH | -40 to +150°C | 0.1 to 40,000 Lux (1 Lux = 0.0079 $\frac{W}{m^2}$) |
| Accuracy | ≤±4%RH @ 25°C, 30 to 80%RH when the power suply is 5 VDC | ±2°C Accuracy from 0°C to +70°C, and -2°C to +6°C Accuracy from -40°C to +150°C | *Not found* |
| Supply Voltage | 5 V DC ±5% | +2.3 to +5.5 V | 2.7 to 3.6 V |
| Current | 0.8 mA (typical) <1.2 mA (maximum) | 6 to 15 $\mu$A | 15 to 500 $\mu$A |
| Operating environment | -40 tp +85°C | -65 to +150°C | -30 to 80°C |
| Responding time | <15 s | <1ms | <13ms |
| stability | <1%RH per year | *Not found* | *Not found* |

$S_a$ = 1.83 GB = $1.83 \times 10^9$ Bytes = $10.83 \times 8 \times 10^9$ bits

where $S_a$ is the actual SD card size.

data rate = 15 kbps = 15000 bit/s (manufacturer provided which is practically achievable).

This yield ($N \times$ packetsize) $< S_a$ when no delay is considered. Therefore, 20 seconds interval is sufficient for accommodating all the data into the 2 GB SD card. Also, the received data (*i.e.*, SD card saved data) is monitored from the control centre through a LabVIEW system. This ensures more reliability and capability for detecting any power failure or other unusual faults of the remote ZigBeePRO–based node. For instance, battery charging status and remaining SD card size are sent with the packet to the control centre. After completion of the data collection, Python program is used for further analysis.

## 3 Result analysis

Stochastic analysis is employed for analysing the result considering 15min averaged-data of both environmental and electrical parameters between 8:30–17:30, which are then evaluated for obtaining linear models. The accuracy and pertinence of the models are determined considering few common but significant statistical indicators, such as $r$, $R^2$, MBE, RMSE, MAPE, and SMAPE. The dimensionless $r$–value has determined the strength of linear relation between environmental and electrical parameters or two environmental parameters in the range of ±1. Another statistical term, $R^2$ has defined the predictive power of the model in connection with the independent parameter. The error terms compute the dispersion of the model's validation results. It is observed that MAPE may cause distortion to the error rate due to the presence of zero or nearly zero data. In such condition, SMAPE performs better measurement than MAPE. The computational formulas of these statistical indicators are shown in Eqs (14)–(16) and (19), Appendix.

### 3.1 Solar irradiance and temperature

Based on individual day data analysis, the lowest irradiance was attained on day1; however, last four hours of the afternoon session of day3 gained less than 200 $\frac{W}{m^2}$ (the least gain among the five days). Therefore, day1 and day3 can be considered as *medium luminance days*. Then, average maximum solar irradiance was noticed on day2 (560 $\frac{W}{m^2}$), day4 (672.94 $\frac{W}{m^2}$), and day5 (663.11 $\frac{W}{m^2}$); so, these three days can be considered as *high luminance days*.

Hourly average (mean) and median solar irradiance data of the five individual days are statistically extracted, analysed and plotted as box–plot in Fig 5. Statistical analysis of the five days has shown that variation of solar irradiance at 8:30 and 17:30 is small. However, the variation is high within this period. The highest variation occurred at 13:30 which is ideally expected to harvest the maximum solar irradiance; however, it did not happen due to the cloudy nature of the days. Hence, considering the statistical hourly median and mean values, the approximate hourly variations of irradiation within the days are shown in the subsequent sections.

Fig 6 shows five days relative humidity with solar irradiance from 8:30 to 17:30. It can be observed that solar irradiance is inversely proportional to relative humidity. From the recorded data, the calculated average peak sun–hour per day is 4.69 hours which corresponds to 16.88 MJ direct solar radiation harvested on average per day. Moreover, the optimum formula for humidity can be obtained in Eq (7). The equation is based on the regression analysis in Fig 6.

$$H_{avg} = -0.04 \cdot Z_{avg} + 69.59 \tag{7}$$

Where, $H_{avg}$ is the average humidity, and $Z_{avg}$ is the measured solar irradiance. The slope,

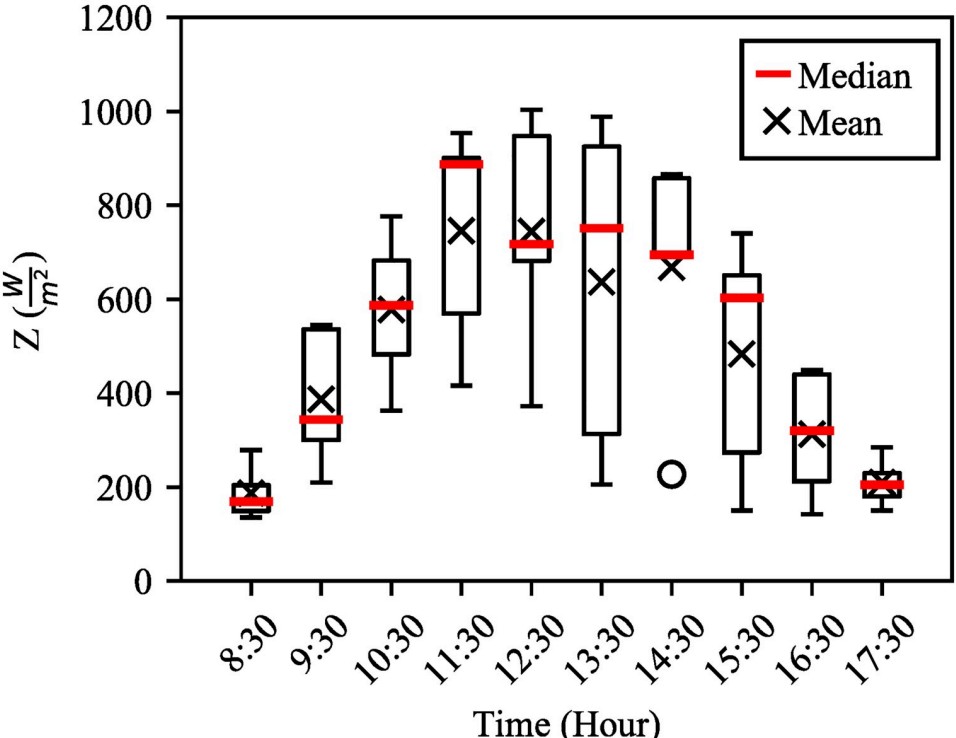

**Fig 5. Statistical analysis of individual day solar irradiance with hourly average.** Red line marker denotes median values at each hour and black (×) marker refers mean value.

-0.04 indicates negative correlation between the solar irradiation and humidity. Further by extracting $\frac{dZ_{avg}}{dH_{avg}}$ from Eq (7), the approximate mathematical model can also be obtained as in Eq (8).

$$\frac{dZ_{avg}}{dH_{avg}} = -25 \qquad (8)$$

Eq (8) states that for every 1% increase in humidity, the solar irradiance drops by 25 $\frac{W}{m^2}$ (the least gain among the five days).

For silicon PV modules, the efficiency is logarithmically dependent on irradiance. Due to that the efficiency was observed almost constant between 200–1000 $\frac{W}{m^2}$. Also, temperature efficiency coefficient is negative, thus efficiency of the both modules goes down with higher irradiance in this outdoor experiment. Fig 7(a)–7(d) shows the effects of module temperature ($T_M$) and solar irradiance ($Z$) on the efficiency ($\eta$) of the a–Si and c–Si modules. It is observed between 11:30 to 13:30 of the day, module efficiency is inversely proportional to the solar irradiance. During this time, hourly average efficiency of c–Si and a–Si are 9.8% and 3.5% respectively at 200–800 $\frac{W}{m^2}$ solar irradiance. c–Si is renowned for higher efficiency and it showed the highest efficiency on both *medium and high luminous days* compared to the a–Si module.

Fig 8 shows five days' average ambient and module temperature. The module temperature of c–Si is about 2.26% higher than a–Si until 11:30 but slightly different (a–Si is 8.1% higher) in the afternoon. However, overall pattern of the module temperature are the same which also

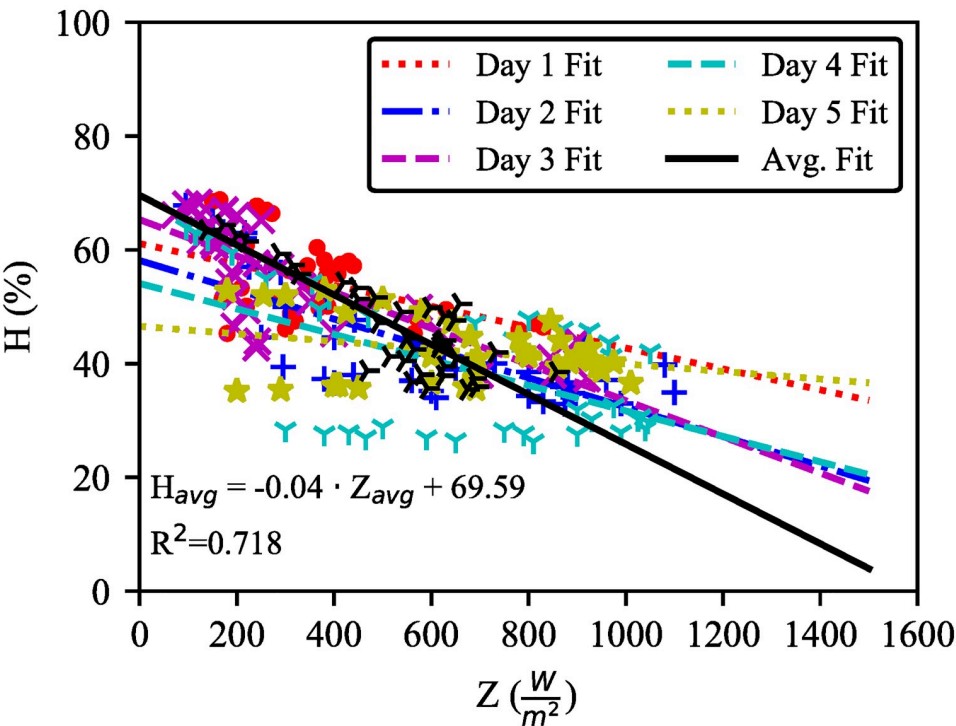

**Fig 6. Inverse proportional relation between relative humidity and solar irradiance.** The mathematical model is fitted to data point with $R^2$ = 0.718. On day1, humidity is between 44.2 and 68.8% with corresponding irradiance of 150–830 $\frac{W}{m^2}$. Day2 is drier than day1 based on humidity (33–67.8%) and irradiance (95–1100 $\frac{W}{m^2}$). Humidity and irradiance on day3 were 35.4–68.3% and 72–920 $\frac{W}{m^2}$ respectively. On day4 (the driest), humidity and irradiance ranges are 25.7–64% and 96–1050 $\frac{W}{m^2}$. Finally, on day5, humidity is observed to be 35–53% when irradiance is 180-1010 $\frac{W}{m^2}$.

reported by [19]. On average, both modules' temperature remained below 58°C (maximum temperature of a–Si and c–Si is 62.9°C and 59.9°C respectively). Comparatively, lower module temperature trend was observed after 13:00 due to continuous 3.88 $\frac{m}{s}$ average wind speed (maximum 5.83 $\frac{m}{s}$ and minimum 0.79 $\frac{m}{s}$) from east to west direction that cools modules' heat. The five days' ambient temperature was between 28.9–34.9°C.

The approximate trend lines of solar irradiance and OPE with corresponding module temperature are shown in Fig 9(a) and 9(b) respectively. Results show that there is a positive correlation among modules temperature, solar irradiance, and OPE. Also, the slope of c–Si is more fitted than a–Si. However, OPE of a–Si is better than c–Si when both modules' temperature is between 30–43°C and solar irradiance is below 500 $\frac{W}{m^2}$). In contrast, solar irradiance above 500 $\frac{W}{m^2}$ and module temperature of 45–53°C, comparatively c–Si performs better. Similar result was found by two researchers on a–Si performance below 400 $\frac{W}{m^2}$ [19] and 600 $\frac{W}{m^2}$ [16]. With this, Fig 10 illustrates the statistical analysis of the five days' efficiencies of a–Si and c–Si modules. A positive correlation is found between the two modules' temperature and the efficiency. However, temperature dependence on the efficiency of a–Si (Fig 10(a)) is not as significant as of c–Si (Fig 10(b)). Fig 10(c) shows that the efficiency of c–Si is almost 50% higher than a–Si at below 48°C module temperature. The rate drops to 36.05% above 48°C.

## 3.2 Module efficiency

The comparison between the effiency of a–Si and c-Si (hourly average) on individual days and five–day average corresponding to daytime is shown in Fig 11. During the high luminance

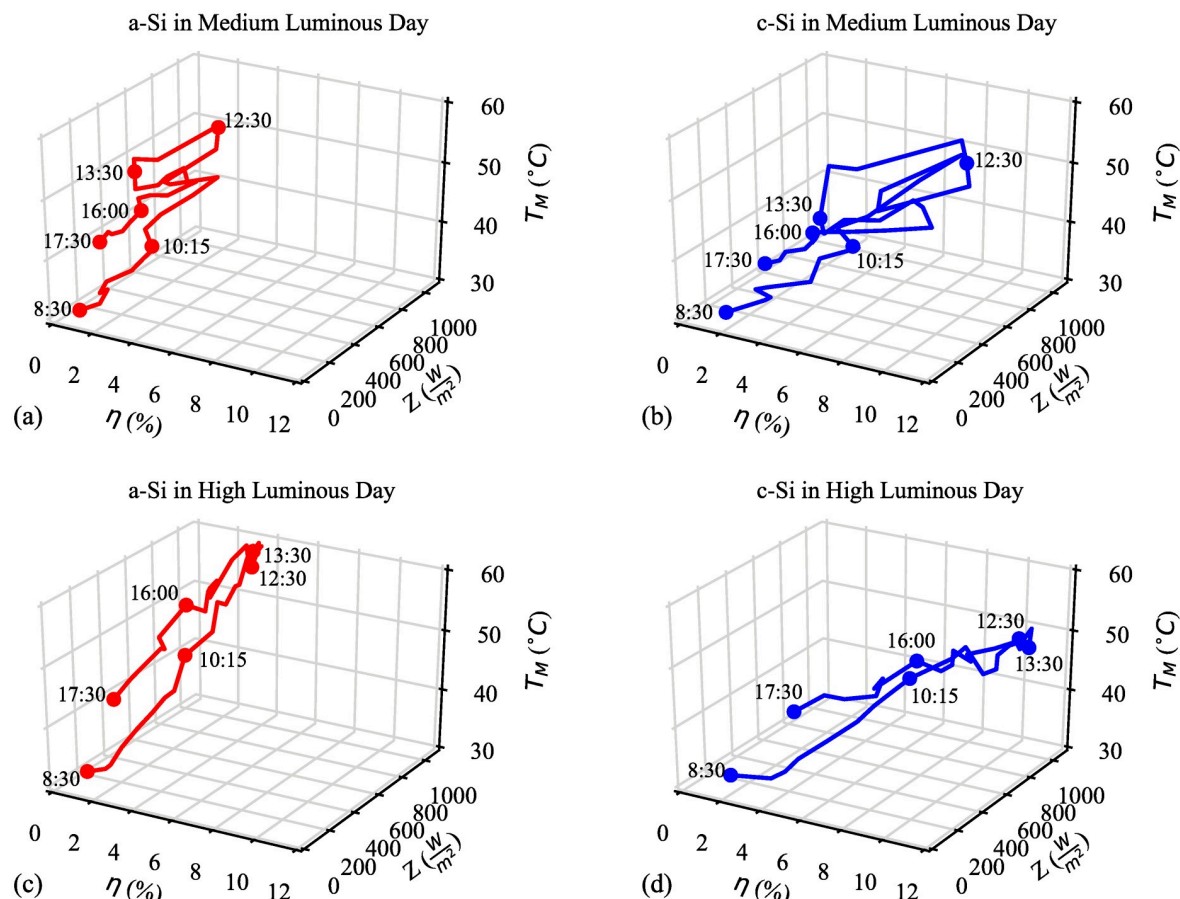

**Fig 7. Effect of module temperature ($T_M$) and solar irradiance ($Z$) on the efficiency ($\eta$) from 8:30 to 17:30 (a)—a–Si and (b) c–Si modules on *medium luminous day*; and (c) a–Si and (d) c–Si modules on *high luminous day*.**

days (day4 and day5), similar trend of efficiency is observed for both modules. The least efficiency was recorded on day3 (medium luminance day); however, a-Si achieved higher efficiency (38.17%) than c-Si (29.92%) as in the manufacturer-rated specification. In Malaysia climate condition, a–Si achieve better efficiency at low irradiance, supported by [15].

Based on the five–day average data, Fig 12 shows that the module efficiency is generally inversely proportional to the solar irradiance. This result is also supported by [19, 47]. The changing rate of solar irradiance shown in Fig 12 explains that the fluctuation of a–Si efficiency is lesser than c–Si efficiency. This is because, a–Si has better shadow tolerance and is less affected by the direction of sunlight. To this extent, $1 \frac{W}{m^2}$ increase in solar irradiance may cause about 0.013% and 0.004% decrease in efficiencies of c–Si and a–Si respectively.

## 3.3 Output Power Efficiency (OPE) and Performance Ratio (PR)

Even though the relationship between OPE and solar irradiance with respect to $R^2$ value is not strong, the $R^2$ value is more significant than a–Si (Fig 13). This is because of the inconsistent distribution of data points for both modules on medium luminance days (day1 and day3). In contrast, PR of a–Si is better than that of c–Si and both modules' PR are inversely proportional to the solar irradiance (Fig 14). The PR is decreased by 14.68% (c–Si) and 24.8% (a–Si)

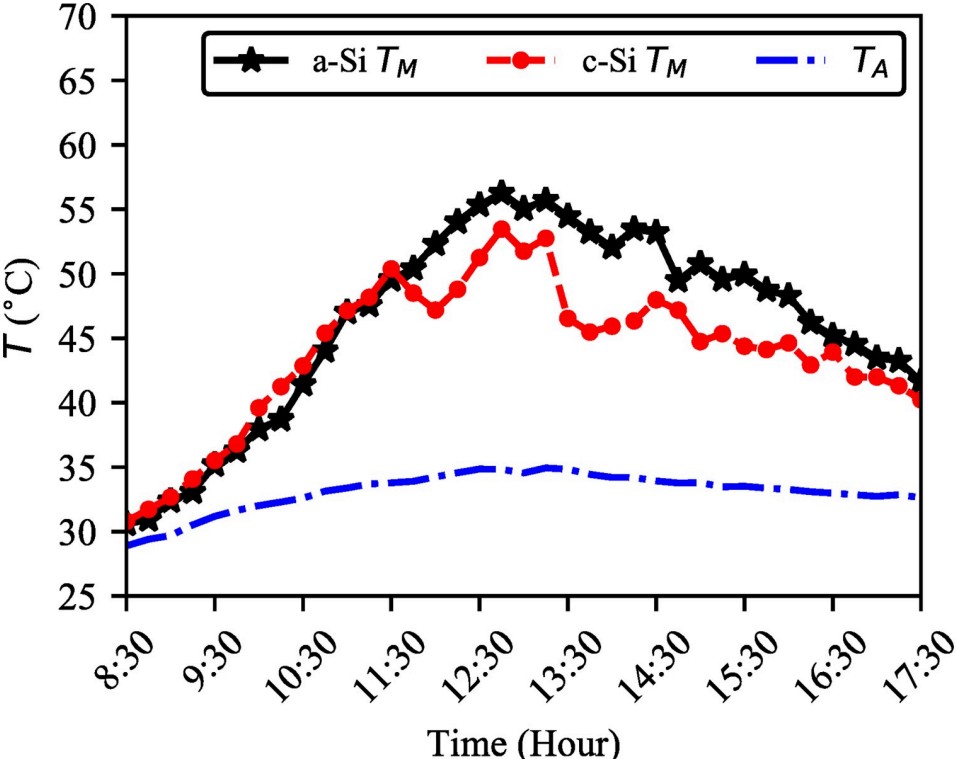

**Fig 8. Comparison between ambient ($T_A$) and the modules' temperature ($T_M$).** Till 11:30, module temperature of c–Si is about 2.26% higher than a–Si. Opposite scenario is seen in the afternoon. The blowing wind maintains the modules' temperature within 58˚C, on average.

between 8:30–12:30 for 298% increase in solar irradiance. This trend is also supported by [19] where the PR was found to be decreased by 5.68% (c–Si) and 22.6% (a–Si) for 175% increase in solar irradiance. The average PR for c–Si and a–Si are 1.02 and 1.21 respectively indicating a–Si for better light absorbing capability during cloudy condition.

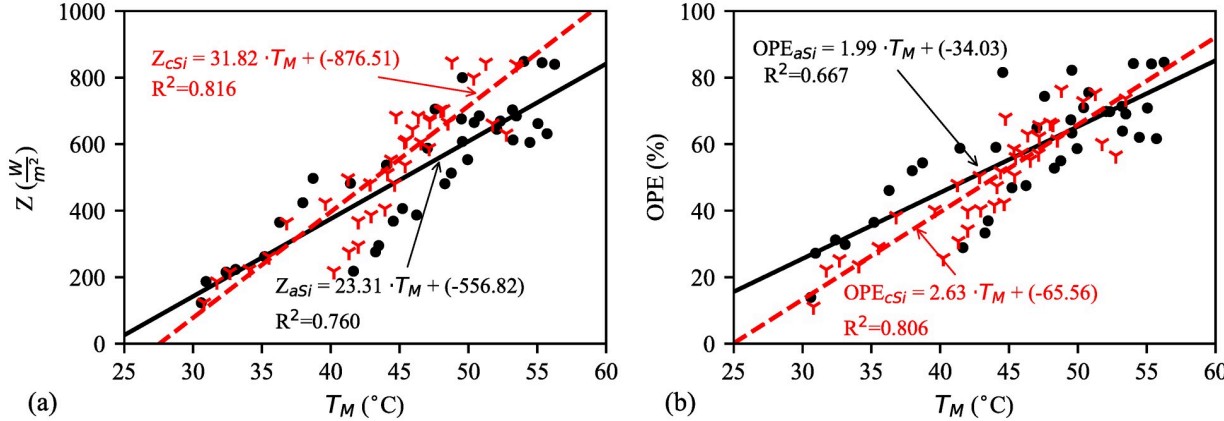

**Fig 9. Comparison between a–Si and c–Si modules' temperature ($T_M$) based on, (a) solar irradiance ($Z$) and (b) output power efficiency (OPE).** $T_M$ is positively correlated with solar irradiance and OPE. By extrapolating the both fitting lines is not valid as it will show modules stop working at 25˚C.

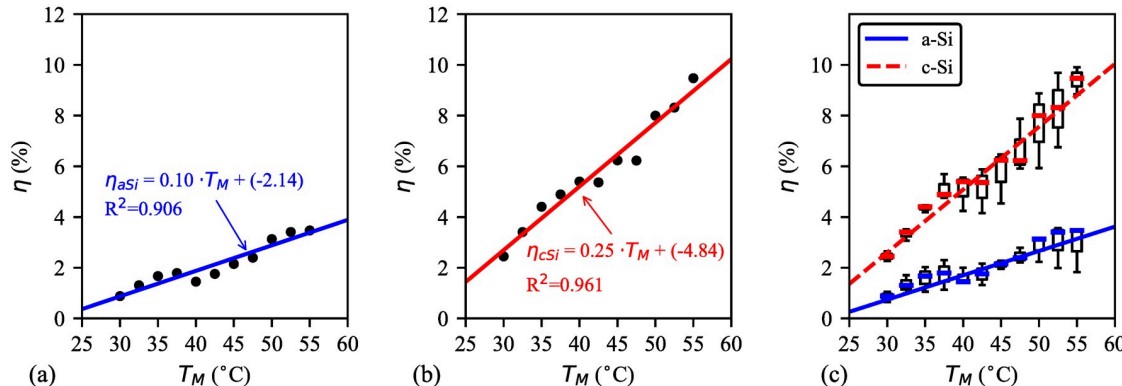

**Fig 10. Statistical analysis of the five days' module temperature and efficiency. (a)** linear trends of a–Si efficiency ($R^2$ = 0.906); **(b)** non–linear trends c–Si efficiency ($R^2$ = 0.961, for linear); **(c)** data deviation for both a–Si and c–Si are along the regression curve.

### 3.4 Statistical analysis

Table 4 shows the regression analysis for the models obtained from the plotted figures. The analysis is also validated by calculating statistical and systematic error terms. The high value of

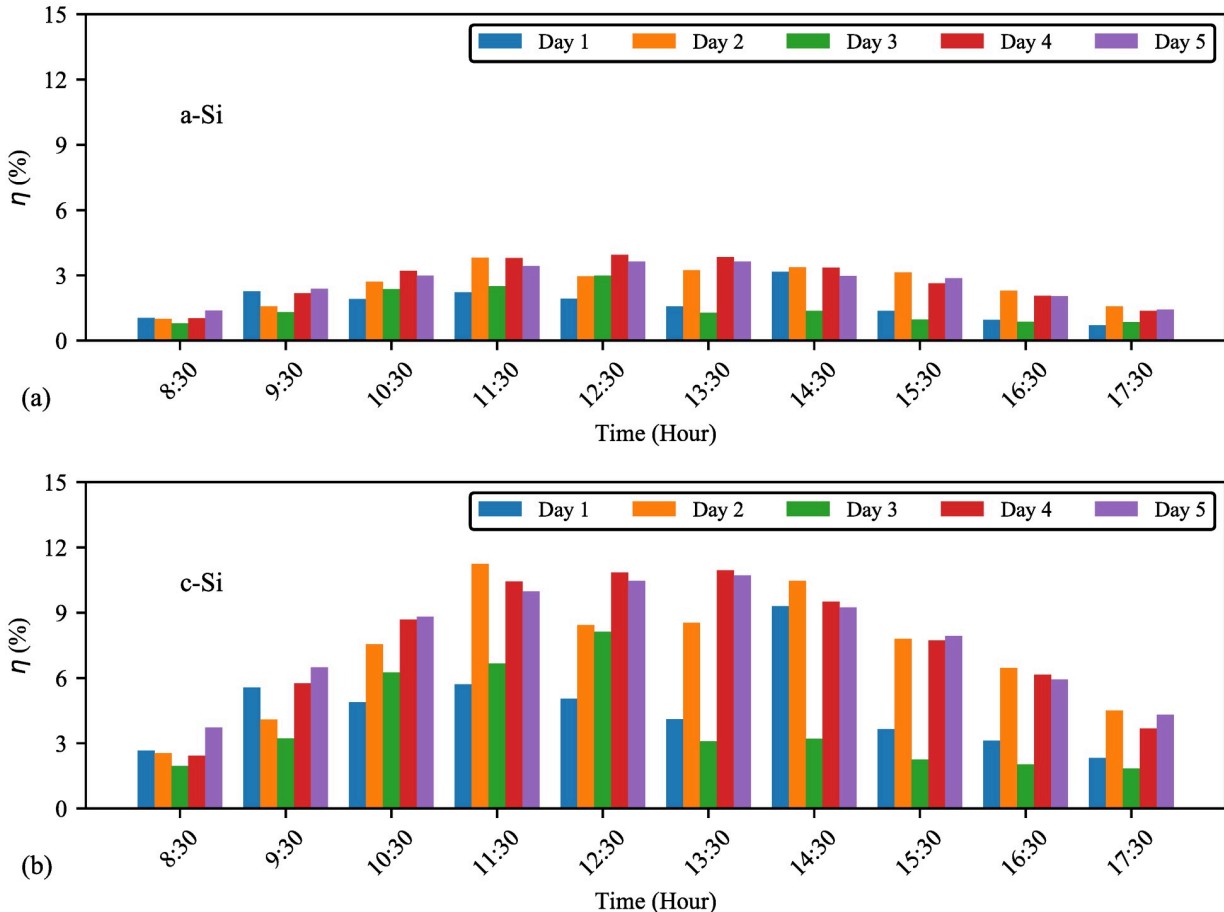

**Fig 11. Comparison of individual days efficiency against daytime (a) a–Si and (b) c–Si.** Similar efficiencies are observed on day4 and day5. Hourly maximum efficiencies of a–Si and c–Si are 3.9% and 11.4% respectively.

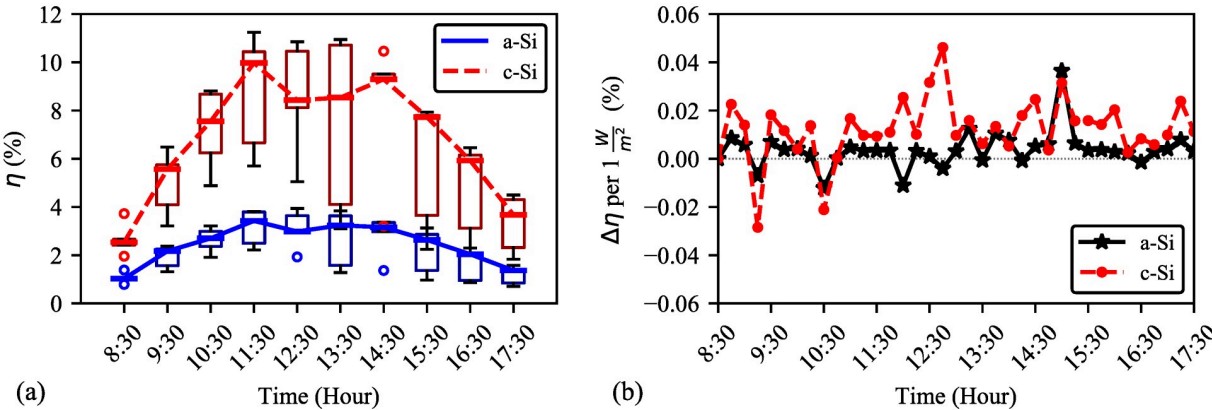

**Fig 12. (a) Five-day average efficiency with solar irradiance.** Maximum, average, and minimum efficiencies are 3.5, 2.3, 0.57% (a–Si) and 9.8, 6.4, 1.4% (c–Si) respectively. **(b) Changes in efficiency with daytime.** Both modules follow similar changing rate of efficiency ($\Delta\eta$ per $1 \frac{W}{m^2}$) against solar irradiance except at 11:30, 12:30, 14:00, and 16:15.

$r$ (0.8168–0.9803) implies that there is a significant relationship between the considered parameters and environmental factor. The accuracy of the models can be further demonstrated by the $R^2$ value outstandingly in Fig 10 compared with the models from Figs 6 and 9(a-Si), which are moderate. Least value of MBE is desirable and it is achieved with acceptable estimation for all the models. Further analysis shows similar observations considering the other error terms, such as RMSE, MAPE, and SMAPE. Besides these indicators, the accuracy of the data can be considered satisfactory based on the calculated t–statistic ($t_s$) which also validate the

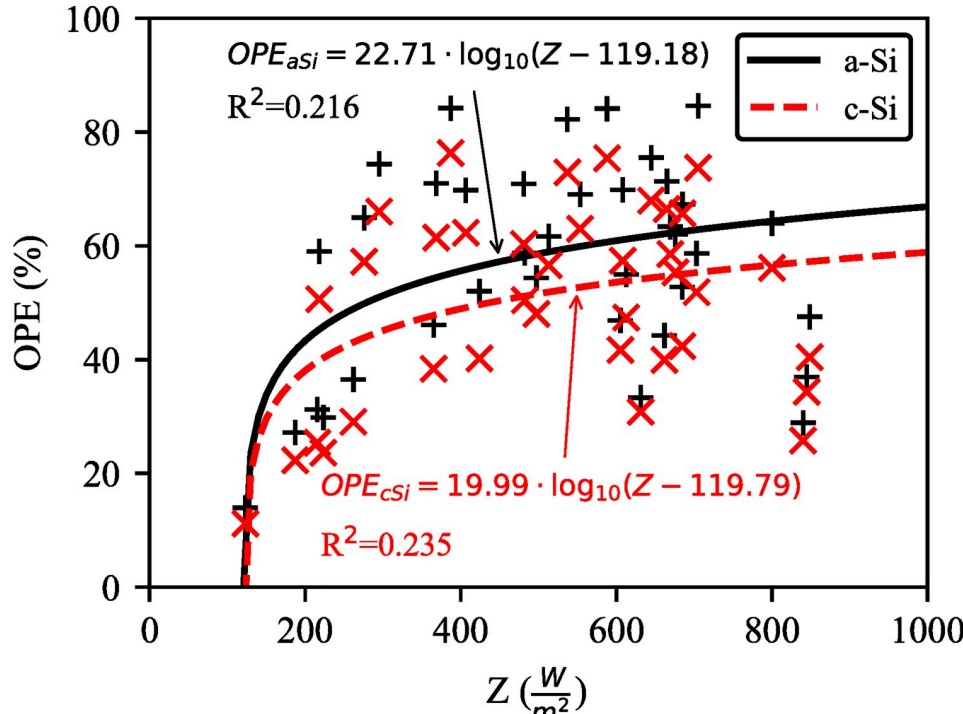

**Fig 13. Five-day average OPE of c–Si and a–Si modules against solar irradiance.** Maximum values of OPE for c–Si and a–Si are 76.33% and 84.60% respectively.

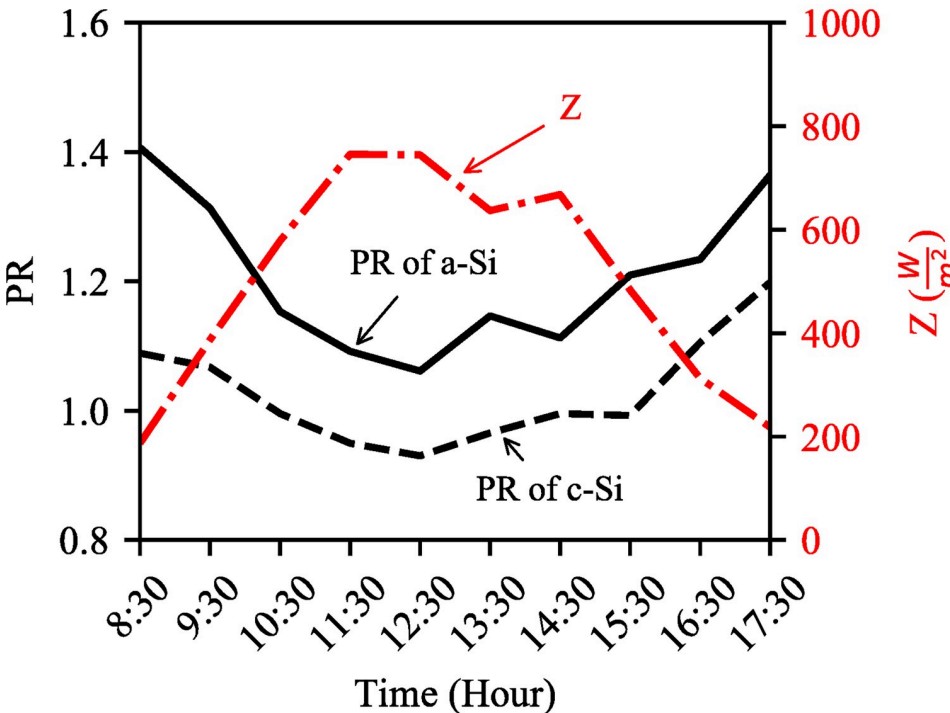

**Fig 14. Hourly average PR of c–Si and a–Si module against daytime and solar irradiance (Z).** The PR and solar irradiance are inversely proportional.

models' estimation in this analysis. For the validation, critical t-statistic ($t_c$) is determined from the standard statistical table considering $(b − 1)$ degree of freedom at 5% significance level with two–tailed test. Here, $b$ is the number of collected data points at every 15 minutes interval from 08:30 to 17:30; therefore, $b$ = (total number of hours × 4 + 1) per day. This $b$ is determined to be 37 for all the models. To ensure the models' estimation with statistical significance, the notation $\{t_s \in \mathbb{R} | -t_c \leqslant t_s \leqslant t_c\}$ has to be true. According to the standard statistical table, the $T_c$ value can be obtained based on $b$ = 37 which confirms the models' validation.

A thorough study from the validation results shows that a–Si or c–Si module efficiency has strong relationship with the module temperature (Fig 10). On the other hand, OPE has strong

**Table 4. Regression analysis of models*** (*e.g.* $Y = \tau \times X + \upsilon$) with validation for UPM, Klang valley region (2.945° North 101.75° East) in Malaysia during dry season.

| Models | | Regression coefficient | | Statistical terms | | | Systematic error terms | | | | Fig. |
|---|---|---|---|---|---|---|---|---|---|---|---|
| (*Y* vs. *X*) | | $\tau$ | $\upsilon$ | $r$ | $R^2$ | $t_s$ | MBE | RMSE | MAPE | SMAPE | |
| *H* vs. *Z* | | -0.0438 | 69.5884 | -0.8473 | 0.7179 | 0.5670 | 0.0105 | 0.1102 | 0.0854 | 0.0839 | Fig 6 |
| *Z* vs. $T_M$ | c-Si | 31.8209 | -876.5052 | 0.9031 | 0.8156 | 0.7033 | 0.0278 | 0.2388 | 0.1625 | 0.1505 | Fig 9(a) |
| | a-Si | 23.3074 | -556.8213 | 0.8720 | 0.7604 | 1.0788 | 0.0455 | 0.2569 | 0.1831 | 0.1693 | |
| OPE vs. $T_M$ | c-Si | 2.6285 | -65.5563 | 0.8977 | 0.8059 | 0.8844 | 0.0268 | 0.1836 | 0.1369 | 0.1302 | Fig 9(b) |
| | a-Si | 1.9864 | -34.0310 | 0.8168 | 0.6672 | 1.2125 | 0.0513 | 0.2587 | 0.1742 | 0.1594 | |
| $\eta$ vs. $T_M$ | c-Si | 0.2511 | -4.8388 | 0.9803 | 0.9611 | 0.1959 | 0.0045 | 0.0733 | 0.0621 | 0.0615 | Fig 10 |
| | a-Si | 0.1005 | -2.1449 | 0.9518 | 0.9059 | 0.2127 | 0.0096 | 0.1432 | 0.1196 | 0.1170 | |

* Each model is expressed as $Y = \tau \times X + \upsilon$ where the regression coefficients ($\tau$, $\upsilon$) and the statistical terms ($r$, $R^2$) are obtained from the respective figures, data analysed by Python program. The systematic error terms and the $t_s$ are calculated from the equations shown in Eq (18), Appendix.

**Table 5. Comparative analysis among STC, experimental data and other researchers' outcomes based on environmental and electrical parameters of c–Si and a–Si module.**

| Parameter | Experimental data | | Previous researchers' outcomes |
|---|---|---|---|
| | c–Si | a–Si | |
| | $A_m = 0.1553\ m^2$ | $A_m = 0.0415\ m^2$ | |
| Environmental parameters: | | | Data shows in order (c–Si, a–Si): |
| $T_{A,avg}$ (˚C) | 33 | | 30.3[†]; 18.1[‡] |
| $Z_{avg}$ ($\frac{W}{m^2}$) | 512.37 | | 625.7[‡]; 593.72[§] |
| $H_{avg}$ (%) | 47 | | 73.4[¶] |
| $Sh$ ($\frac{h}{day}$) | 4.69 | | |
| Wind speed (m/s) | 3.88 E→W | | 5.5[¶] |
| $T_{M,avg}$ (˚C) | 43.7 | 46 | 40.22, 39.14[†]; 28.5,27.2[‡] |
| Electrical Parameters: | | | |
| $I_{avg}$ (mA) | 632.95 | 64.85 | |
| $I_{STC}$ (mA) | 1140 | 100 | |
| $V_{avg}$ (V) | 15.7 | 15.22 | |
| $V_{STC}$ (V) | 17.4 | 17 | |
| $P_{avg}$ (W) | 9.94 | 0.99 | |
| $E_{avg}$ ($\frac{Wh}{day}$) | 46.62 | 4.64 | |
| $P_{STC}$ (W) | 20 | 1.7 | |
| $FF_{avg}$ | 0.51 | 0.51 | 0.712, 0.56[†] |
| $\eta_{avg}$ (%) | 6.4 | 2.39 | 6.87, 2.23[†]; 13.1, 5.5[‡]; c–Si 9.53[¶] |
| $\eta_{STC}$ (%) | 12.9 | 4 | 4.4, 2.16[§] |
| $OPE_{avg}$ (%) | 49.7 | 58.23 | 33.1, 33.74[†]; 52, 55.5[‡]; 22.885, 14.71[§] |
| $PR_{avg}$ | 0.97 | 1.14 | 0.933, 1.046[†]; 0.85, 1.03[‡]; 44.81, 28.8[§] |
| Yield factor ($kWh/kWp/day$) | 2.33 | 2.73 | 1.41, 1.58[†]; 2.44, 2.60[‡] |

[†] [15]

[‡] [19]

[§] [16]

[¶] [18].

relationship with the modules temperature (Fig 9). The relation between humidity and solar irradiance is not strong (Fig 6), which means that other factors, such as degree of cloud cover, ambient temperature, atmospheric dust, and water vapour density weaken the relationship. The validation results are evident for the similar climate characteristic during dry season in Malaysia.

In Table 5, a set of PV measurement parameters are considered for comparing STC, experimental results and other researchers' outcome. We have observed that the environmental parameter, such as average ambient temperature is similar to the research [15], which leads to approximately similar outcome in FF, module efficiency, PR, and yield factor; however, OPE is varied. The result is also closely matched with that of [19] in terms of FF, OPE, PR, and yield factor. Since both researches did not have peak sun-hour data, we obtained their yield factor by considering similar peak sun-hour of this study, 4.69 $\frac{h}{day}$. The yield factors of this study for c-Si and a-Si are found to be 2.33 and 2.73 $\frac{kWh}{kWp}$ respectively, which are also within 2.6 ±0.15 $\frac{kWh}{kWp}$, referred by [48]. In case of efficiency, the researcher, found higher outcome than this study due to variance in ambient temperature [19]. The average maximum powers for a–Si and c–Si achieved in this study are 1.63 W and 17.45 W respectively and are found to be 4.12% (a–Si)

and 12.75% (c–Si) less than STC rated values. In comparison with the average module efficiency, c–Si and a–Si attained 49.6% and 59.75% respectively of their STC rated-value. Moreover, a–Si shows better performance over c–Si in the case of the other three parameters, OPE PR, and yield factor.

## 3.5 Estimation of energy yield for NEM

Based on the experimental analysis on *medium and high luminance days*, the total energy yield in kWh during dry season can be estimated. For the estimation, a 1 kW capacity of c–Si and a–Si PV modules are assumed for modeling purpose in accordance with Malaysia's NEM application. The modeling equation for estimating the total energy yield, $E$ in the unit of kWh, during dry season is derived as in Eq (9).

$$E = E_1 + E_2 \tag{9}$$

where,

$$E_1 = \sum_{i=0}^{|\alpha D|} P_i \cdot Sh_i \tag{10}$$

and

$$E_2 = \sum_{j=0}^{|\beta D|significance=1} P_j \cdot Sh_j \tag{11}$$

Here, $\alpha$ and $\beta$ are probability of *medium and high luminance days* respectively. Therefore, $\alpha + \beta = 1$ and $0 \leq (\alpha, \beta) \leq 1$. In Eqs (10) and (11), $P_i$ and $P_j$ denote total output power in $\alpha D$ and $\beta D$ days respectively. $Sh_i$ and $Sh_j$ are sun–hour, and $D$ denotes the total number of days in dry season.

To validate the estimated model, we have considered other models from [49] and [50] shown in Eqs (12) and (13) respectively.

$$P = \eta_{STC} \cdot A_M \cdot Z_a [1 - \epsilon_1 \cdot (T_M - 25)] \tag{12}$$

$$P = \eta_{STC} \cdot A_M \cdot Z_a [1 - \epsilon_2 \cdot (T_M - 25)] \tag{13}$$

In Eqs (12) and (13), the value of temperature coefficient, $\epsilon_1$ is taken from c–Si and a–Si module datasheet specified as $0.0045\degree C^{-1}$ and $0.0020\degree C^{-1}$ respectively. $\epsilon_2$ is also considered $0.0044\degree C^{-1}$ (c–Si) and $0.0026\degree C^{-1}$ (a–Si) by [50]. The module area ($A_M$) for 1 kW capacity is calculated as 7.765 $m^2$ for c–Si and 24.41 $m^2$ for a–Si. Finally, the energy yields are shown in Table 6. The data is normalized for comparison based on the experimental output power in kWh when the capacities of both modules are 1 kW and sizes are 7.765 $m^2$ (c–Si) and 24.41 $m^2$ (a–Si).

Statistically, if 70% and 30% of days during dry season are considered as *medium and high luminance day* respectively, the total energy output of a–Si (125.23 kWh) is higher than that of c–Si (108.85 kWh) based on our estimation model, Eq (9). Similar energy output can be noticed for the models Eqs (12) and (13). The percentage difference between our model (Eq (9)) and that of Eq (12) is 6.55% (c–Si) and 17.01% (a–Si) respectively. On the other hand, the difference is 6.19% (c–Si) and 18.18% (a–Si) considering model Eq (13). This insignificant difference validates our model, deemed appropriate for the NEM.

**Table 6. Energy yield in kWh estimation during dry season (Jun–Jul) for NEM application in Malaysia.**

| Module type | $Z_{a,avg}$ $(\frac{W}{m^2})$ | $T_{M,avg}$ (°C) | $P$ $(\frac{W}{day})$ | | | $sh$ (h) | $E$ (kWh), Eq (9) | | | | | |
|---|---|---|---|---|---|---|---|---|---|---|---|---|
| | | | | | | | Model [50] | | Model [49] | | Our model | |
| | | | [50], Eq (13) | [49], Eq (12) | Experimental | | $E_1$ | $E_2$ | $E_1$ | $E_2$ | $E_1$ | $E_2$ |
| $\alpha = 0.7$ and $D = 61$. So, 42 days are *medium luminance* | | | | | | | | | | | | |
| c–Si | 340.64 | 41.85 | 315.92 | 313.92 | 342.83 | 3.07 | 40.73 | – | 40.48 | – | 44.21 | – |
| a–Si | 340.64 | 41.08 | 318.03 | 321.9 | 412.44 | 3.07 | 41 | – | 41.51 | – | 53.18 | – |
| $\beta = 0.3$ and $D = 61$. So, 19 days are *high luminance* | | | | | | | | | | | | |
| c–Si | 626.86 | 44.96 | 572.77 | 571.52 | 603.18 | 5.64 | – | 61.38 | – | 61.24 | – | 64.64 |
| a–Si | 626.86 | 49.22 | 573.52 | 582.42 | 672.36 | 5.64 | – | 61.46 | – | 62.41 | – | 72.05 |

Total energy yield, $E$ (kWh):

From our estimation model, Eq (9): 108.85 (c–Si) and 125.24 (a–Si)

From [49] Eq (12): 101.72 (c–Si) and 103.92 (a–Si)

From [50] Eq (13): 102.11 (c–Si) and 102.46 (a–Si)

$Z_{avg,a}$ refers to the experimental average solar irradiance on either *medium luminance* (day1, day3) or *high luminance* (day2, day4, day5) days. Similarly, $T_{M,avg}$ and $Sh$ show the module temperature and sun–hour on *medium and high luminance days* respectively. The output power, $P$ (W/day) are calculated from Eqs (12) and (13) for 1 kW module, size 7.765 m$^2$ (c–Si) and 24.41m2 (a–Si). In contrast, the experimental $P$ (W/day) for 1 kW module is normalized from the actual data and average of *medium and high luminance days'*. The actual average output power of c–Si and a–Si are respectively 342.8289 W and 412.4377 W on *medium luminance day*; whereas 603.1826 and 672.4411 W on *high luminance day*. Total energy yield is calculated based on Eq (9) which shows higher energy output of a–Si (125.24 kWh) compared to c–Si (108.85 kWh). Similar trends are found for Eqs (12) and (13). This comparison analysis validates our estimation model.

Based on the real data and model analysis, it is observed that total energy output of a–Si is 15.07% higher than c–Si, which is optimum between these recommended two modules during dry season in Malaysia.

## 4 Conclusion

In this study, performance evaluation of a–Si and c–Si PV modules and their dry–season energy yield prediction model are developed for NEM in Malaysia. The evidence from this study confirms that 4.69 h of average peak sun–hour, minimum humidity value of 25.7%, and maximum solar irradiance of $1100 \frac{W}{m^2}$ are achievable at Klang valley of peninsular Malaysia during the dry season. However, the environmental data monitored by self–developed wireless smart system shows that $38.9 \frac{W}{m^2}$ solar irradiance can be dropped with 1% increase in humidity. Module temperatures of both the modules do not exceed 58°C due to the blowing wind at $3.88 \frac{m}{s}$, on average. From the evaluation of electrical parameters of the modules, it is observed that the average efficiencies attained about 49.6% (c–Si) and 59.75% (a–Si) of the manufacturer rated efficiency. This attainment has occurred due to module temperature significant to c–Si which causes 13.95% less efficient when the module temperature exceeded 48°C. Also, a–Si has achieved better OPE than c–Si at $< 500 \frac{W}{m^2}$ solar irradiance and between 30-43°C module temperature, whereas opposite performance is noticed above $500 \frac{W}{m^2}$ solar irradiance. Due to better light absorbing capability during cloudy condition, the average PR of a–Si (1.21) is higher than c–Si (1.02). The PR is found to be inversely proportional to the solar irradiance and thus, decreased by 14.68% (c–Si) and 24.8% (a–Si) with 298% increase in solar irradiance. In addition, the yield factor of a-Si (2.73 $\frac{kWh}{kWp}$) is found to be higher than c–Si (2.33 $\frac{kWh}{kWp}$). The regression analysis validates most of the obtained models based on electrical and environmental parameters by confirming the statistical and systematic error terms. The strongest (module efficiency versus module temperature) and the weakest (OPE versus module temperature)

relations are determined by calculating $r = 0.9803$, $R^2 = 0.9611$ and $r = 0.8168$, $R^2 = 0.6672$ from the respective models. Based on the results of the evaluation, the proposed model estimated total energy yield in kWh during the dry season for the NEM monthly reimbursement. The model projects that if 70% is *medium* and 30% is *high luminance days* of the dry season, a–Si produces 15.07% more energy than c–Si. The overall information suggests promoting a–Si module due to its higher energy yield, PR, OPE, yield factor, and cost over c–Si when the size of the module is compromised. Future research should therefore focus on the investigation of better–performed PV module during the secondary maximum and maximum rainfall seasons. Thus, it can determine the total energy yield in kWh of all the seasons in order to come up to a decision about feasibility, right choices of modules, and the fastest payback of NEM investment in Malaysia.

## Appendix

*Computation formula for statistical error terms*

The formulas shown in Eqs (14) to (18) are:

$$\mathrm{MBE} = \frac{1}{b} \cdot \sum_{i=0}^{b} \frac{\phi_{e_i} - \phi_{a_i}}{\phi_{a_i}} \tag{14}$$

$$\mathrm{RMSE} = \sqrt{\frac{1}{b} \cdot \sum_{i=0}^{b} \left( \frac{\phi_{e_i} - \phi_{a_i}}{\phi_{a_i}} \right)^2} \tag{15}$$

$$\mathrm{MAPE} = \frac{1}{b} \cdot \sum_{i=0}^{b} \left| \frac{\phi_{a_i} - \phi_{e_i}}{\phi_{a_i}} \right| \tag{16}$$

$$\mathrm{SMAPE} = \frac{2}{b} \cdot \sum_{i=0}^{b} \frac{|\phi_{e_i} - \phi_{a_i}|}{\phi_{e_i} + \phi_{a_i}} \tag{17}$$

$$t_s = \sqrt{\frac{(b-1) \cdot \mathrm{MBE}^2}{\mathrm{RMSE}^2 - \mathrm{MBE}^2}} \tag{18}$$

where, $b$ is the number of collected data at every 15 minutes interval from 08:30 to 17:30; therefore $b =$ (total hours $\cdot$ 4 + 1) per day. $\phi_{e_i}$ and $\phi_{a_i}$ represent estimated and actual data respectively. While comparing the $X$ with $Y$ axis actual data samples, correlation coefficient ($r$) is evaluated by using the following expression (19):

$$\text{Correlation coefficient } (r) = \frac{b \cdot \sum_{i=0}^{b} X_i \cdot Y_i - \left( \sum_{i=0}^{b} X_i \cdot \sum_{i=0}^{b} Y_i \right)}{\sqrt{b \sum_{i=0}^{b} (X_i)^2 - \left( \sum_{i=0}^{b} X_i \right)^2} \cdot \sqrt{b \sum_{i=0}^{b} (Y_i)^2 - \left( \sum_{i=0}^{b} Y_i \right)^2}} \tag{19}$$

The value of $r$ is $-1 \leq r \leq +1$ and closest to $-1$ or $+1$ indicates perfect negative or positive fit respectively. The negative ($-$) or positive ($+$) sign denotes relationship between $X_i$ and $Y_i$ such that by increasing $X_i$, $Y_i$ decreases or increasing $X_i$, $Y_i$ also increases, *i.e.*

$r > 0$ refers to positive linear relationship between $X_i$ and $Y_i$.

$r < 0$ refers to negative linear relationship between $X_i$ and $Y_i$.

$r = 0$ refers to weak or no linear relationship between $X_i$ and $Y_i$.

## Acknowledgments

The authors would like to acknowledge the Malaysia Meteorological Department for support with data in this study.

## Author Contributions

**Conceptualization:** Syed Zahurul Islam, Muhammad Saufi, Arash Toudeshki, Syed Zahidul Islam.

**Data curation:** Syed Zahurul Islam, Arash Toudeshki.

**Formal analysis:** Syed Zahurul Islam, Syed Zahidul Islam.

**Funding acquisition:** Mohammad Lutfi Othman, Muhammad Saufi, Rosli Omar.

**Investigation:** Syed Zahidul Islam.

**Methodology:** Syed Zahurul Islam, Arash Toudeshki.

**Project administration:** Mohammad Lutfi Othman, Muhammad Saufi.

**Resources:** Mohammad Lutfi Othman, Muhammad Saufi, Rosli Omar.

**Software:** Syed Zahurul Islam, Arash Toudeshki.

**Supervision:** Mohammad Lutfi Othman.

**Validation:** Syed Zahurul Islam, Arash Toudeshki.

**Visualization:** Arash Toudeshki.

**Writing – original draft:** Syed Zahurul Islam, Rosli Omar.

**Writing – review & editing:** Mohammad Lutfi Othman, Muhammad Saufi, Syed Zahidul Islam.

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
