## [Decision Letter · Decision Letter 0]

30 Jul 2020

PONE-D-20-15047

Photovoltaic Modules Evaluation and Dry-Season Energy Yield Prediction Model for NEM in Malaysia

PLOS ONE

Dear Dr. Syed Islam,

Thank you for submitting your manuscript to PLOS ONE. After careful consideration, we feel that it has merit but does not fully meet PLOS ONE’s publication criteria as it currently stands. Therefore, we invite you to submit a revised version of the manuscript that addresses the points raised during the review process.

As queries raised by the reviewers appended below (and my own evaluation of the manuscript), the manuscript should be carefully revised or rearranged to clearly demonstrate its novelty specifically the abstract, introduction and conclusion parts of the manuscript. After careful revision of the manuscript, the manuscript will be sent out to peer reviews again to evaluate its quality for further acceptance in the PLOS ONE journal. 

We look forward to receiving your revised manuscript.

Kind regards,

Mahesh Suryawanshi, Ph. D.

Academic Editor

PLOS ONE

Journal Requirements:

2.We suggest you thoroughly copyedit your manuscript for language usage, spelling, and grammar. If you do not know anyone who can help you do this, you may wish to consider employing a professional scientific editing service.  

3.In your Data Availability statement, you have not specified where the minimal data set underlying the results described in your manuscript can be found. PLOS defines a study's minimal data set as the underlying data used to reach the conclusions drawn in the manuscript and any additional data required to replicate the reported study findings in their entirety. All PLOS journals require that the minimal data set be made fully available. For more information about our data policy, please see http://journals.plos.org/plosone/s/data-availability.

4.Thank you for stating the following in the Financial Disclosure section:

[1. Zahurul Syed, Research Management Center, Research Fund E15501, Universiti Tun Hussein Onn Malaysia (UTHM), www.uthm.edu.my.

2. Mohammad Lutfi, 9671700, Geran Putra Berimpak, University Putra Malaysia (UPM), www.upm.edu.my

The funders had no role in study design, data collection and analysis, decision to publish, or preparation of the manuscript.].   

We note that one or more of the authors are employed by a commercial company: Radiation Solutions Inc,

5. We note you have included a table to which you do not refer in the text of your manuscript. Please ensure that you refer to Table 5 in your text; if accepted, production will need this reference to link the reader to the Table.

Reviewers' comments:

Reviewer's Responses to Questions

**Comments to the Author**

1. Is the manuscript technically sound, and do the data support the conclusions?

Reviewer #1: Yes

Reviewer #2: Partly

2. Has the statistical analysis been performed appropriately and rigorously? 

Reviewer #1: Yes

Reviewer #2: Yes

3. Have the authors made all data underlying the findings in their manuscript fully available?

Reviewer #1: Yes

Reviewer #2: No

4. Is the manuscript presented in an intelligible fashion and written in standard English?

Reviewer #1: No

Reviewer #2: Yes

5. Review Comments to the Author

Reviewer #1: The manuscript is reviewed carefully, widely statistical analysis is given in this study for a small period.

In my opinion a small period is not enough for showing the differences between types.

The whole manuscript has to be revised carefully,

Some common symbols, for temperature, for efficiency (nu nopt ro) has to be used.

please use c or a for crystalline silicon and amorphous silicon in small letters.

please check the sentence in line 41 (there is a dimension problem)

Also check line 45.

It is suggested to combine sentence with previous one (line 67) because a reference is necessary.

check and open the sentence on line 68 and explain the reason.

Please explain which 3 days on line 80.

It is suggested not to start a new sentence with a ref. number, In X, Table X or Fig. X. but cite them in the sentence.

It is suggested to remove "and environmental" from the heading of section 2.

It is more convenient to compare the energy rating (kWh/kWp) and it is suggested to write in the manuscript.

The conc. section is supported by data It is good.

Reviewer #2: 1. I suggest reorganizing the abstract, highlighting the novelties introduced, it should contain answers to the following questions:

• What problem was studied and why is it important?

• What methods were used?

• What are the important results?

• What conclusions can be drawn from the results?

• What is the novelty of the work and where does it go beyond previous efforts in the literature?

The originality of the paper needs to be stated clearly.

---

## [Author Response · Author response to Decision Letter 0]

11 Sep 2020

We refer to the ‘Response to Reviewers’ for the detail of our actions against each comment.

---

## [Decision Letter · Decision Letter 1]

23 Oct 2020

Photovoltaic Modules Evaluation and Dry-Season Energy Yield Prediction Model for NEM in Malaysia

PONE-D-20-15047R1

Dear Dr. Syed Zahurul Islam,

We’re pleased to inform you that your manuscript has been judged scientifically suitable for publication and will be formally accepted for publication once it meets all outstanding technical requirements.

Kind regards,

Mahesh Suryawanshi, Ph. D.

Academic Editor

PLOS ONE

Additional Editor Comments (optional):

Reviewers' comments:

Reviewer's Responses to Questions

**Comments to the Author**

1. If the authors have adequately addressed your comments raised in a previous round of review and you feel that this manuscript is now acceptable for publication, you may indicate that here to bypass the “Comments to the Author” section, enter your conflict of interest statement in the “Confidential to Editor” section, and submit your "Accept" recommendation.

Reviewer #1: All comments have been addressed

2. Is the manuscript technically sound, and do the data support the conclusions?

Reviewer #1: Yes

3. Has the statistical analysis been performed appropriately and rigorously? 

Reviewer #1: Yes

4. Have the authors made all data underlying the findings in their manuscript fully available?

Reviewer #1: Yes

5. Is the manuscript presented in an intelligible fashion and written in standard English?

Reviewer #1: Yes

6. Review Comments to the Author

Reviewer #1: It is understood that the manuscript is revised and It is better in this format.

The authors revise the manuscript according to the suggestions.

7. PLOS authors have the option to publish the peer review history of their article (what does this mean?). If published, this will include your full peer review and any attached files.

Reviewer #1: **Yes**

---

## [Editor Report · Acceptance letter]

30 Oct 2020

PONE-D-20-15047R1 

Photovoltaic Modules Evaluation and Dry-Season Energy Yield Prediction Model for NEM in Malaysia 

Dear Dr. Islam:

I'm pleased to inform you that your manuscript has been deemed suitable for publication in PLOS ONE. Congratulations! Your manuscript is now with our production department. 

Kind regards, 

on behalf of

Dr. Mahesh Suryawanshi 

Academic Editor

PLOS ONE